# Positional Encoder Graph Quantile Neural Networks for Geographic Data

**William E. R. de Amorim**[*]                                      *william__rappel@hotmail.com*
*Department of Statistics*
*University of Brasília*
*Brazil*

**Scott A. Sisson**                                                 *scott.sisson@unsw.edu.au*
*School of Mathematics and Statistics*
*University of New South Wales*
*Australia*

**Thais C. V. Rodrigues**                                           *thaisrodrigues@unb.br*
*Department of Statistics*
*University of Brasília*
*Brazil*

**David J. Nott**                                                   *standj@nus.edu.sg*
*Department of Statistics and Data Science*
*National University of Singapore*
*Singapore*

**Guilherme S. Rodrigues**                                          *guilhermerodrigues@unb.br*
*Department of Statistics*
*University of Brasília*
*Brazil*

**Reviewed on OpenReview:** `https://openreview.net/forum?id=5PjL8ZOPBt`

## Abstract

Positional Encoder Graph Neural Networks (PE-GNNs) are among the most effective models for learning from continuous spatial data. However, their predictive distributions are often poorly calibrated, limiting their utility in applications that require reliable uncertainty quantification. We propose the Positional Encoder Graph Quantile Neural Network (PE-GQNN), a novel framework that combines PE-GNNs with Quantile Neural Networks, partially monotonic neural blocks, and post-hoc recalibration techniques. The PE-GQNN enables flexible and robust conditional density estimation with minimal assumptions about the target distribution, and it extends naturally to tasks beyond spatial data. Empirical results on benchmark datasets show that the PE-GQNN outperforms existing methods in both predictive accuracy and uncertainty quantification, without incurring additional computational cost. We also identify important special cases arising from our formulation, including the PE-GNN.

---

[*]Corresponding author.

# 1 Introduction

Large spatial datasets are naturally generated in a wide range of applications in economics (Anselin, 2022), meteorology (Bi et al., 2023), urban transportation (Lv et al., 2014; Derrow-Pinion et al., 2021; Kashyap et al., 2022), social networks (Xu et al., 2020), e-commerce (Sreenivasa & Nirmala, 2019) and other fields. Gaussian Processes (GPs) (Rasmussen & Williams, 2006; Cressie & Wikle, 2011) are a fundamental tool for modelling spatial data on continuous domains. They are flexible and interpretable models for unknown functions, both in spatial and more general regression settings. However, with time complexity $O(n^3)$ and storage complexity $O(n^2)$, naive GP methods quickly become intractable for large datasets (Liu et al., 2019). This has led to a large range of approximate inference methods, such as those based on sparse approximations to covariance or precision matrices (Furrer et al., 2006; Lindgren et al., 2011), low rank approximations (Cressie et al., 2022) or nearest neighbour approximations (Vecchia, 1998; Datta et al., 2016; Katzfuss & Guinness, 2021).

Given the difficulty of GP computations, it is of interest to explore scalable methods for large spatial datasets using neural networks (NNs) and to enhance their ability to quantify uncertainty. A prominent method for making spatial predictions using Graph Neural Networks (GNNs) is the Positional Encoder Graph Neural Network (PE-GNN) of Klemmer et al. (2023). Our contribution is to make key modifications to the PE-GNN architecture to enhance its ability to make accurate spatial predictions and to better quantify uncertainty. These modifications will be explained further below.

NNs are popular in data modeling and prediction tasks like computer vision and natural language processing (NLP). However, traditional NNs struggle to handle spatial dynamics or graph-based data effectively. GNNs (Kipf & Welling, 2017; Veličković et al., 2018; Hamilton et al., 2017) offer a powerful and scalable method for applying NNs to graph-structured data. The idea is to share information through the edges of a graph, allowing nodes to exchange information during learning. GNNs are versatile and can uncover non-linear relationships among inputs, hidden layers, and each node's neighborhood information. The success of GNNs in spatial applications largely depends on the spatial graph construction, including choice of distance metric and the number of neighboring nodes, and traditional GNNs often struggle to model complex spatial relationships. To address this, Klemmer et al. (2023) introduced the PE-GNN, which enhances predictive performance in spatial interpolation and regression. However, the PE-GNN is not designed to provide a full probabilistic description of the target's distribution, and assuming a Gaussian distribution for predictions can lead to poorly calibrated intervals, such as 80% intervals that fail to contain the true outcome 80% of the time (for a more detailed discussion of calibration definitions, see Kock et al., 2024). Recently, Bao et al. (2024) proposed a new framework called Spatial Multi-Attention Conditional Neural Processes (SMACNPs) for spatial small sample prediction tasks. SMACNPs use GPs parameterized by NNs to predict the target variable distribution, which enables precise predictions while quantifying the uncertainty of these predictions. However, these methods remain constrained to producing Gaussian predictive distributions, limiting their capacity to represent asymmetric or multimodal random variates.

Methods based on quantile regression are an alternative approach to probabilistic forecasting making rapid progress in recent years. Tagasovska & Lopez-Paz (2019) introduced the Simultaneous Quantile Regression (SQR) loss function that we use in our formulation. Si et al. (2022) proposed a novel architecture for estimating generic quantiles of a conditional distribution. In one dimension, this method produces a quantile function regression $\hat{q}(\tau)|\boldsymbol{x}$ that estimates the $\tau$-th quantile of the predictive distribution of the target variable given a feature vector $\boldsymbol{x}$. Kuleshov & Deshpande (2022) argue that the method of Si et al. (2022) is inefficient with high-dimensional predictors. To address this, they modify the original formulation to incorporate a post hoc recalibration procedure whereby an auxiliary model recalibrates the predictions of a trained model. The first model outputs features, usually summary statistics like quantiles, representing a low-dimensional view of the conditional distribution. The auxiliary model, the recalibrator, uses these features as input to produce calibrated predictions using Si *et al.*'s quantile function regression framework. The main drawback is that it requires training two separate models, each needing its own training set.

Our work makes two sets of contributions. (1) We propose a new architecture that merges the two-step procedure of Kuleshov & Deshpande (2022) into a single model by postponing the concatenation of the $\tau$ value used by Si et al. (2022). Additionally, $\tau$ is transformed beforehand to facilitate learning and provide

better control over the form of the predictive distribution. To ensure the model outputs valid, non-crossing quantile functions, we employ the partially monotonic blocks introduced by Nolte et al. (2023). This approach enhances the network's ability to model uncertainty and the model becomes more robust to high-dimensional predictor spaces. (2) We introduce structural changes to the PE-GNN. Instead of applying the GNN operator to the concatenation of the nodes' features and the spatial embedding, we apply it only to the features. In addition, we introduce the mean target value of a node's neighbours as a feature after the GNN layers, closer to the output.

The structure of this work is as follows: Section 2 offers a brief background overview, Section 3 outlines the proposed method for geographic data prediction, Section 4 shows experimental results on four real-world datasets, and Section 5 concludes.

## 2 Background

**Graph Neural Networks:** Graph Neural Networks (GNNs) are powerful and scalable tools for representation learning and inference on graph-structured data. They exploit the topological relationships between adjacent nodes to produce context-aware embeddings, which can be effectively used in downstream tasks (Wu et al., 2022). GNN layers iteratively refine each node's embedding by aggregating information from its own features as well as those of its neighboring nodes.

Graph Convolutional Networks (GCNs) (Kipf & Welling, 2017) are a specific type of GNN layer inspired by the convolution operations used in Convolutional Neural Networks (CNNs). For weighted graphs, a GCN layer $k$ updates node embeddings according to the following equation:

$$\boldsymbol{H}^{(k)} = f^{(k)}\left(\boldsymbol{D}^{-1/2}\left[\boldsymbol{A} + \boldsymbol{I}\right]\boldsymbol{D}^{-1/2}\boldsymbol{H}^{(k-1)}\boldsymbol{W}^{(k)}\right), \quad \text{for } k \in \{1, \ldots, K\}. \tag{1}$$

In this formulation, the input to the network is the feature matrix $\boldsymbol{H}^{(0)} = \boldsymbol{X}$. The function $f^{(k)}$ denotes a non-linear activation function (e.g., ReLU), and $\boldsymbol{W}^{(k)}$ is a learnable weight matrix. The adjacency matrix $\boldsymbol{A}$ encodes the graph structure, with edge weights typically derived from node distances and zero entries for unconnected pairs. The identity matrix $\boldsymbol{I}$ adds self-loops, and $\boldsymbol{D}$ is the corresponding *degree* matrix. In our experiments (Section 4), we also consider other widely used GNN architectures, namely, Graph Attention Networks (GATs) (Veličković et al., 2018) and GraphSAGE (Hamilton et al., 2017).

**Positional Encoder Graph Neural Network:** In a typical spatial regression setting, each datapoint is represented as $p_i = \{y_i, \boldsymbol{x}_i, \boldsymbol{c}_i\}$, where $y_i$ is a continuous scalar target variable, $\boldsymbol{x}_i$ is a vector of input features, and $\boldsymbol{c}_i$ denotes the geographical coordinates associated with observation $i$. A given batch of datapoints $B = \{p_1, \ldots, p_{n_B}\}$ can be fully represented by three matrices: the target vector $\boldsymbol{y}_B \in \mathbb{R}^{n_B \times 1}$, the feature matrix $\boldsymbol{X}_B \in \mathbb{R}^{n_B \times p}$, and the coordinate matrix $\boldsymbol{C}_B \in \mathbb{R}^{n_B \times 2}$, respectively.

Klemmer et al. (2023) introduced a novel approach for incorporating spatial information into GNNs: the Positional Encoder Graph Neural Network (PE-GNN). In this framework, a positional encoder (PE) processes the spatial coordinate matrix $\boldsymbol{C}_B$ to produce a learned spatial embedding matrix $\boldsymbol{C}_B^{\text{emb}}$. The embedding is computed as $\boldsymbol{C}_B^{\text{emb}} = PE(\boldsymbol{C}_B, \sigma_{\min}, \sigma_{\max}, \Theta_{\text{PE}}) = NN(ST(\boldsymbol{C}_B, \sigma_{\min}, \sigma_{\max}), \Theta_{\text{PE}})$, where $ST$ denotes a deterministic set of sinusoidal transformations with hyperparameters $\sigma_{\min}$ and $\sigma_{\max}$, and $NN$ is a fully connected neural network with trainable parameters $\Theta_{\text{PE}}$. A complete description of the PE mechanism is provided in Appendix A.2.

The matrix $\boldsymbol{C}_B^{\text{emb}}$ is concatenated column-wise with the node features before the application of GNN layers. Thus, the input to the first GNN layer is given by $\boldsymbol{H}_B^{(0)} = \text{concat}(\boldsymbol{X}_B, \boldsymbol{C}_B^{\text{emb}})$. At each training step, a new random batch of nodes $B$ is sampled, and the full pipeline — graph construction, spatial embedding generation, feature concatenation, and GNN propagation — is executed using only the nodes in that batch. For each node $p_i \in \{p_1, \ldots, p_{n_B}\}$, the PE-GNN predicts a target value $\hat{y}_i$ and, as an auxiliary task (Klemmer & Neill, 2021), the corresponding *Local Moran's I* statistic (Anselin, 1995), denoted by $\hat{I}(y_i)$. The total loss used by Klemmer et al. (2023) combines both objectives:

$$\mathcal{L}_B = \text{MSE}(\hat{\boldsymbol{y}}_B, \boldsymbol{y}_B) + \lambda \, \text{MSE}(\hat{I}(\boldsymbol{y}_B), I(\boldsymbol{y}_B)),$$

where $\lambda$ controls the contribution of the auxiliary task.

**Quantile regression:** Koenker & Bassett Jr (1978) proposed a linear quantile regression model to estimate conditional distribution quantiles. It uses the pinball loss $\rho_\tau(r_i) = \max\left(\tau r_i, (\tau - 1)r_i\right)$, where $r_i = y_i - \hat{q}_i(\tau)$, $\hat{q}_i(\tau) = \boldsymbol{X}_i\hat{\boldsymbol{\beta}}$, and $\tau$ is the desired cumulative probability associated with the predicted quantile $\hat{q}_i(\tau)$. The pinball loss for the $i$-th observation is $\rho_\tau(r_i)$. The loss over a dataset is the average $\rho_\tau(r_i)$ value over all datapoints. A natural extension of quantile linear regression is quantile neural networks (QNNs). This approach is illustrated in Figure 4a, which seeks to estimate the conditional quantiles for a pre-defined grid $(\tau^1, \ldots, \tau^d)$. Each quantile is estimated by an independent model (Figure 4a). This can lead to quantile predictions with quantile crossing (e.g., a median prediction lower than the first quartile prediction). Rodrigues & Pereira (2020) proposed an approach that outputs multiple predictions: one for the expectation and one for each quantile of interest. The loss function is:

$$\mathcal{L} = \frac{1}{d+1} \left[ \text{MSE}\left(\hat{\boldsymbol{y}}, \boldsymbol{y}\right) + \sum_{i=1}^{n} \sum_{j=1}^{d} \frac{\rho_{\tau^j}\left(y_i - \hat{q}_i(\tau^j)\right)}{n} \right]. \tag{2}$$

Tagasovska & Lopez-Paz (2019) proposed a method to generate a model that is independent of quantile selection. During training, for each datapoint in the batch, a Monte Carlo sample $\tau \sim U(0, 1)$ is drawn and concatenated with the corresponding datapoint feature vector. The SQR loss function is similar to Eqn. 2, but they predict random quantiles $\mathcal{L} = \frac{1}{n} \sum_{i=1}^{n} \rho_{\tau_i}(y_i - \hat{q}_i(\tau_i))$, with $\{\tau_i\}_1^n \sim U(0, 1)$. As the network learns, it becomes able to provide a direct estimate to *any* quantile of interest. Hence, this procedure outputs an inherently calibrated model suitable for conditional density estimation. Si et al. (2022) construct NNs with a similar loss function (Figure 4c).

Kuleshov & Deshpande (2022) adapted the architecture from Si et al. (2022) into a two-step process for larger predictor spaces (Figure 4d). First, a model is trained to take the original features as inputs and generate low-dimensional representations of the predicted distribution. Next, a recalibrator is trained using these *new* features by minimizing the estimated expected pinball loss over $\tau$. During inference, the recalibrator takes the new features and an arbitrary $\tau$ as inputs to produce the quantile prediction. This method is highly dependent on the choice of recalibrator features.

## 3 Method

In this work, we introduce the **Positional Encoder Graph Quantile Neural Network (PE-GQNN)**, a novel framework for predictive modeling on spatial data. Algorithm 1 shows the step-by-step procedure to train a **PE-GQNN** model.

Figure 1 illustrates its complete pipeline. Here, each rectangle labeled "GNN", "LINEAR" and "MONO-TONIC" represents a set of one or more neural network layers, with the type of each layer defined by the title inside the rectangle. At each layer, a nonlinear transformation (e.g. ReLU) may be applied.

After initializing the model and hyperparameters, the first step of **PE-GQNN** is to randomly sample a batch $B$ of datapoints. The next step projects the matrix of geographical coordinates $\boldsymbol{C}_B$ into the positional embeddings, $\boldsymbol{C}_B^{emb}{}_{(n_B \times u)}$ (Algorithm 1, Step 5). $\boldsymbol{C}_B$ is also used to compute the distance between each pair of datapoints (Step 6). From these distances and a predefined number of nearest neighbors, a graph can be constructed, with each datapoint as a node and edge weights computed from the distances, leading to the batch adjacency matrix $\boldsymbol{A}_B$.

At Step 13, the first distinction between **PE-GQNN** and PE-GNN arises: instead of using the concatenation of the feature matrix and the spatial embedding as the input for the GNN operator, we apply the GNN operator only to the feature matrix $\boldsymbol{X}_B$. One or more fully-connected layers are then used to reduce the feature embedding dimensionality. This process receives the constructed graph and the batch feature matrix $\boldsymbol{X}_{B(n_B \times p)}$ as input and yields an embedding matrix of features as output: $\boldsymbol{X}_B^{emb}{}_{(n_B \times g)}$. Step 14 performs a column concatenation between the feature embedding $\boldsymbol{X}_B^{emb}{}_{(n_B \times g)}$ and the output obtained from the PE: $\boldsymbol{C}_B^{emb}{}_{(n_B \times u)}$. This concatenation results in the matrix $\boldsymbol{L}_{B(n_B \times (g+u))}$.

---

**Algorithm 1** PE-GQNN training

---

**Require:**

Training data target, features, and coordinates matrices: $\boldsymbol{y}_{(n \times 1)}$, $\boldsymbol{X}_{(n \times p)}$, and $\boldsymbol{C}_{(n \times 2)}$.

A positive integer $k$ defining the number of neighbors considered in the spatial graph.

Positive integers $tsteps$ and $n_B$, the number of training steps and the batch size.

Positive integers $u$, $g$, and $s$, the embedding dimensions considered in, respectively, the PE, the GNN layers, and the layer where we introduce $\boldsymbol{\tau}$ and $\bar{\boldsymbol{y}}$.

An activation function $f()$ for $\boldsymbol{\tau}$.

**Ensure:**

A set of learned weights for the model initialized at Step 1.

1: Initialize model with random weights and hyperparameters.
2: Set optimizer with hyperparameters.

3: **for** $b \leftarrow 1$ **to** $tsteps$ **do**                                      ▷ Batched training
4:      Sample minibatch $B$ of $n_B$ datapoints: $\boldsymbol{X}_{B(n_B \times p)}$, $\boldsymbol{C}_{B(n_B \times 2)}$, $\boldsymbol{y}_{B(n_B \times 1)}$.
5:      Input $\boldsymbol{C}_{B(n_B \times 2)}$ into PE, which outputs the batch's spatial embedding matrix $\boldsymbol{C}_B^{emb}{}_{(n_B \times u)}$.
6:      Compute the great-circle distance between each pair of datapoints from $\boldsymbol{C}_B$.
7:      Construct a graph using $k$-nearest neighbors from the distances computed in Step 6.
8:      Set $\boldsymbol{A}_B$ as the adjacency matrix of the graph constructed in Step 7.
9:      **for** $i \leftarrow 1$ **to** $n_B$ **do**
10:          Using $\boldsymbol{A}_B$, compute $\bar{y}_i = \frac{1}{k} \sum_{j=1}^{k} y_j$, where $j = 1, \ldots, k$ are the neighbors of $i$.
11:      **end for**
12:      Set $\bar{\boldsymbol{y}}_B = [\bar{y}_1, \ldots, \bar{y}_{n_B}]^\top$.
13:      Apply GNN layers to the features $\boldsymbol{X}_{B(n_B \times p)}$, followed by fully-connected layers to reduce dimensionality. This step outputs a feature embedding matrix $\boldsymbol{X}_B^{emb}{}_{(n_B \times g)}$.
14:      Column concatenate $\boldsymbol{X}_B^{emb}{}_{(n_B \times g)}$ with $\boldsymbol{C}_B^{emb}{}_{(n_B \times u)}$, which results in $\boldsymbol{L}_{B(n_B \times (g+u))}$.
15:      Apply fully-connected layers to reduce $\boldsymbol{L}_{B(n_B \times (g+u))}$ to $\boldsymbol{\phi}_{B(n_B \times s)}$.
16:      Create a vector with values sampled from $U(0,1)$: $\boldsymbol{\tau}_{B(n_B \times 1)} = [\tau_1, \ldots, \tau_{n_B}]^\top$.
17:      Column concatenate $\boldsymbol{\phi}_B$ with $f(\boldsymbol{\tau}_B)$ and $\bar{\boldsymbol{y}}_B$ to create $\widetilde{\boldsymbol{\phi}}_{B(n_B \times (s+2))}$.
18:      Predict the target quantile vector $[\hat{q}_1(\tau_1), \ldots, \hat{q}_{n_B}(\tau_{n_B})]^\top$ using $\widetilde{\boldsymbol{\phi}}_B$.
19:      Compute loss $\mathcal{L}_B = \frac{1}{n_B} \sum_{i=1}^{n_B} \rho_{\tau_i}(y_i - \hat{q}_i(\tau_i))$.
20:      Update the parameters of the model using stochastic gradient descent.
21: **end for**

---

Subsequently, we use one or more fully-connected layers (Step 15) to reduce the dimensionality of $\boldsymbol{L}_B$, making it suitable for two innovations in **PE-GQNN**. This set of fully-connected layers outputs the matrix $\boldsymbol{\phi}_{B(n_B \times s)}$, which is then combined with $\bar{\boldsymbol{y}}_B$ and $\boldsymbol{\tau}_B$. $\bar{\boldsymbol{y}}_B$ represents a vector with one scalar for each datapoint in the batch, containing the mean target variable among the training neighbours for each node. It is computed using the graph constructed in previous steps (Step 10), and has dimensions $n_B \times 1$. It is comparable to a vector of predictions generated by a KNN regression model, where neighbours are determined using the distance calculated from geographical coordinates. Here, we used the simple average due to its relationship with KNN prediction; however, one could use a weighted average via the adjacency matrix $\boldsymbol{A}_B$. We introduce this input at a later stage to avoid data leakage. If the GNN operator received $\bar{\boldsymbol{y}}_B$ as input, after completing the message passing process in each GNN layer, the true node target value would inadvertently be transmitted to its neighbours, creating potential data leakage (Appleby et al., 2020).

In the same layer where $\bar{\boldsymbol{y}}_B$ is introduced, we apply a similar approach to Si et al. (2022) to make **PE-GQNN** an inherently calibrated model suitable for probabilistic and quantile predictions. For each batch $B$, we create a $n_B \times 1$ vector $\boldsymbol{\tau}_{B(n_B \times 1)} = [\tau_1, \ldots, \tau_{n_B}]^\top$ of random $U(0,1)$ draws (Step 16). Then, we column concatenate $\boldsymbol{\phi}_B$ with $f(\boldsymbol{\tau}_B)$ and $\bar{\boldsymbol{y}}_B$ to create $\widetilde{\boldsymbol{\phi}}_{B(n_B \times (s+2))}$ (Step 17), where $f()$ is an activation function. Next, forward propagation is computed (Step 18) in one or more fully-connected layers, outputting predicted

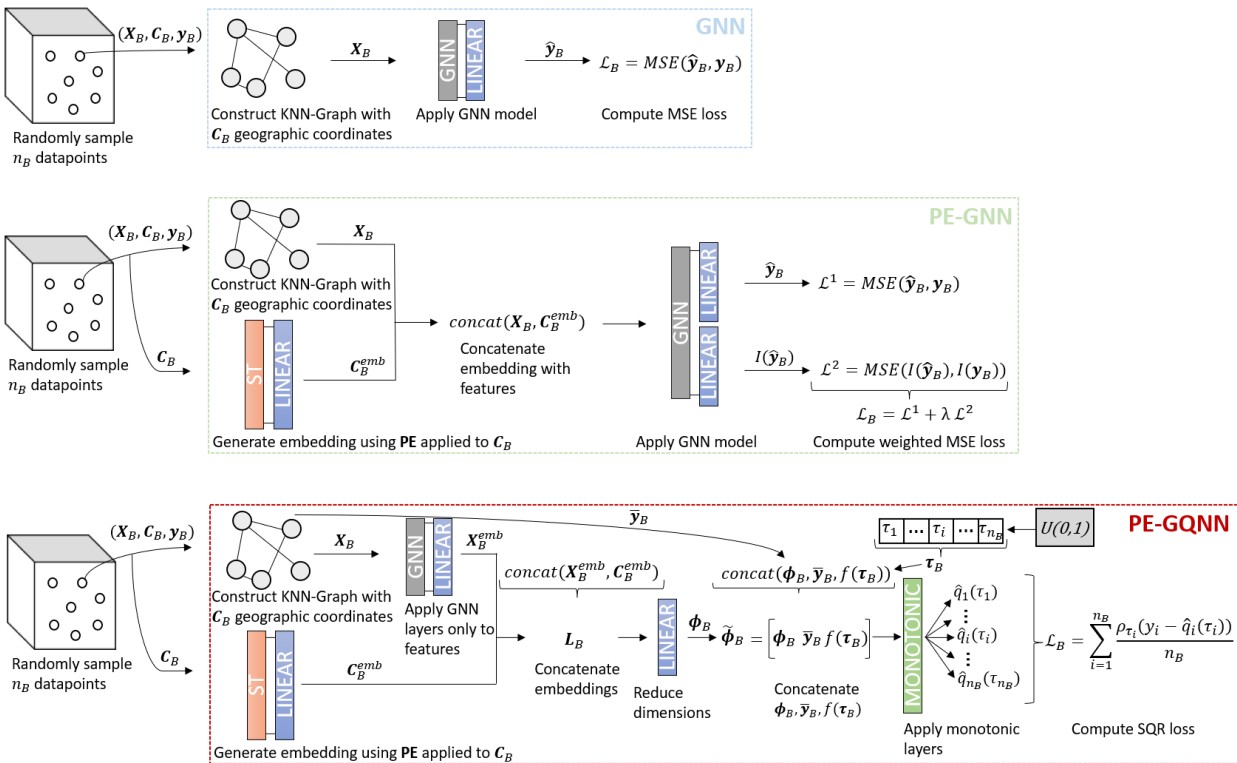

Figure 1: **PE-GQNN** compared to PE-GNN and GNN

quantiles for each datapoint in the batch. These predictions are then used to compute the SQR loss function introduced by Tagasovska & Lopez-Paz (2019).

Incorporating $\tau$ values into the model architecture improves its ability to model uncertainty and serves as a regularization mechanism (Rodrigues & Pereira, 2020). The use of SQR loss acts as a natural regularizer, producing a detailed description of the predictive density beyond just mean and variance estimation. For predictions, the quantile of interest, $\tau$, must be given, along with the basic data components (e.g. $\tau = 0.25$ gives the first quartile). If interest is in predicting multiple quantiles for the same observation, the input can be propagated up to the layer where $\tau$ is introduced. For each quantile of interest, propagation can be limited to the final layers.

**Homoscedastic Gaussian model as a particular case:** If in Steps 17 and 18 one sets $f(\tau) = \Phi^{-1}(\tau)$, where $\Phi()$ denotes the PDF of a standard Gaussian distribution, and use a single linear predictive layer, the model's quantile function,

$$\hat{q}_i(\tau) = (b + w_{\bar{y}_i}\bar{y}_i + \sum_{j=1}^{s} w_j\phi_j) + w_\tau\Phi^{-1}(\tau) = \mu_i + \sigma\Phi^{-1}(\tau), \ \forall i \in 1, \ldots, n_B, \tag{3}$$

matches the quantile function of a Gaussian random variable, therefore reducing **PE-GQNN** to a homoscedastic Gaussian regression model with mean $\mu_i$ and a common, learnable standard deviation $\sigma = w_\tau$. Here, $b$, $w_\tau$, $w_{\bar{y}_i}$, and $\{w_j\}$ are the prediction layer parameters, and $\{\phi_j\}$ are the activation values from the previous layer. This result may appear counterintuitive, as the SQR loss can produce a model structurally identical to one trained with a fundamentally different objective, namely, the MSE loss. Similarly, if $f(\tau) = \text{logit}(\tau)$ and a single linear predictive layer is employed, the model yields logistic predictive distributions, a continuous distribution that should not be confused with logistic regression. Therefore, a well-chosen activation function $f(\tau)$ not only facilitates learning but also explicitly shapes the structure of the predictive distributions. The use of a set of monotonic layers enhances the flexibility of our approach, significantly reducing the reliance on the choice of $f(\tau)$.

**Quantile crossing and Monotonic Blocks:** Several related approaches (Tagasovska & Lopez-Paz, 2019; Rodrigues & Pereira, 2020; Si et al., 2022; Kuleshov & Deshpande, 2022) are subject to quantile crossing, a phenomenon that occurs when the requirement that higher quantiles be greater than or equal to lower quantiles is violated. To ensure the network outputs valid probability distributions (i.e. $\hat{q}(\tau, \widetilde{\phi}, \bar{y}) \leq \hat{q}(\tau', \widetilde{\phi}, \bar{y}), \forall \tau < \tau'$), we propose using the Lipschitz Monotonic Networks (LMN) introduced by Nolte et al. (2023) to approximate $q(\tau, \widetilde{\phi}, \bar{y})$. Those are highly expressive blocks of layers that can approximate all monotonic Lipschitz bounded functions. That is, any function for which there exists a constant $\lambda$ such that $|\hat{q}(\tau, \widetilde{\phi}, \bar{y}) - \hat{q}(\tau', \widetilde{\phi}, \bar{y})| \leq \lambda|\tau - \tau'|, \forall 0 < \tau, \tau' < 1$.

By design, there is a trade-off between model expressiveness and probabilistic coherence. The decision to include monotonic layers should be guided by the specific application. In practice, it is straightforward to assess on a validation set whether quantile crossing is significant. When it is not (especially since SQR (Tagasovska & Lopez-Paz, 2019) already mitigates this issue), adding monotonic layers may be unnecessary or even counterproductive. As usual, we recommend exploring multiple network configurations during hyperparameter tuning to achieve a good balance between expressiveness and coherence.

**Number of Monte Carlo samples:** When applying the framework proposed by Si et al. (2022), we followed Tagasovska & Lopez-Paz (2019) and set $d = 1$ for the $\tau$ values. Let $\mathcal{L}(\theta, \tau, \boldsymbol{x}, y)$ denote the loss function for a given quantile $\tau \sim \mathrm{U}(0, 1)$ and an observed pair $(\boldsymbol{x}, y) \sim D_{\mathrm{data}}$, where $D_{\mathrm{data}}$ represents the full data-generating process. At each training iteration, we minimize the empirical loss $\mathcal{L}_B$, which, by the Law of Large Numbers, converges to $\tilde{\mathcal{L}}(\theta) = \mathbb{E}\tau, \boldsymbol{x}, y\mathcal{L}(\theta, \tau, \boldsymbol{x}, y)$ as the batch size $n_B \to \infty$. Consequently, the gradient estimates converge to the same limit for any $d$. Since the batch size in our experiments is sufficiently large (see Section 4.2), each gradient estimate effectively aggregates losses over more than a thousand distinct quantile levels. Thus, setting ($d = 1$) simplifies the implementation without compromising performance, as demonstrated in Section 4.

**Target domain:** The final layer should preferably use an activation function coherent with the domain of the target variable, ensuring model outputs are valid for target distribution support. E.g., an exponential function could be appropriate if the target variable is continuous, unbounded and positive (Goodfellow et al., 2016).

# 4 Experiments

## 4.1 Experimental setup

The **PE-GQNN** was implemented using PyTorch (Paszke et al., 2019) and PyTorch Geometric (Fey & Lenssen, 2019). The source code is available at: `https://github.com/WilliamRappel98/PE-GQNN`. We conducted comprehensive simulations to explore the prediction performance and other properties of the proposed model. Computation was performed on an Intel i7 (7th Generation) processor.

**Candidate models:** The experiment was designed to compare five primary approaches for addressing spatial regression problems across four diverse real-world datasets (California Housing, Air Temperature, 3D Road, and Australian Census). See Table 4. Table 1 lists each candidate model and their applicable datasets. All models were trained using the Adam optimizer (Kingma & Ba, 2015), with early stopping employed to prevent overfitting. For GNN-based approaches, we used $k = 5$ nearest neighbors to construct the graphs. The learning rate was set to 0.001 across all models. A batch size of 1,024 was used for the Air Temperature dataset, while a batch size of 2,048 was used for the remaining three datasets. The architectural details are given in Appendix Section A.3.

Approach I involves the traditional application of GNNs to geographic data. Three types of GNN layers were considered: GCNs (Kipf & Welling, 2017), GATs (Veličković et al., 2018), and GSAGE (Hamilton et al., 2017). For each of these, the architecture remains consistent to facilitate performance comparisons: two GCN/GAT/GSAGE layers with ReLU activation and dropout, followed by a linear prediction layer.

Table 1: Summary of candidate models.

| Approach | Model | Type | PE | Innovations | Loss | Datasets |
|----------|-------|------|-----|-------------|------|----------|
| I | GNN | GNN | No | No | MSE | All |
| II | PE-GNN $\lambda = $ best | GNN | Yes | No | $\text{MSE}_y + \lambda\text{MSE}_{I(y)}$ | All |
| III | PE-GNN (with SQR) | GNN | Yes | Partial | SQR | California |
| IV | PE-GQNN | GNN | Yes | Yes | SQR | All |
| V | SMACNP | GP | No | No | $-$ Log Likelihood | California and Air Temp. |

Approach II involves the application of PE-GNN (Klemmer et al., 2023) with optimal weights for each dataset and layer type combination, as demonstrated by the experimental findings of Klemmer et al. (2023). The GNN architecture used is the same as for approach I. The code is available at github[1].

Approach III represents a naive combination of the PE-GNN with the quantile regression framework described in Section 3. Specifically, we trained the PE-GNN with the SQR loss function, concatenating $f(\tau) = \tau$ immediately after the GNN layers. Approach IV, which is the primary focus of this research, is the **PE-GQNN**. Compared to Approach III, it introduces the following innovations: $\tau$ is incorporated into the network through $\text{probit}(\tau)$, but only after reducing $L_B$ into $\phi_B$. The GNN layers no longer process the positional encoders, and the final fully connected layers are replaced with monotonic blocks. Additionally, $\bar{y}$ is used as a feature in the final part of the network. The architectures of the PE and GNN layers remain identical to those in the previous approaches.

Finally, a benchmark approach that does not use GNNs but was recently proposed for modelling spatial data will be considered as approach V: SMACNPs. This approach, proposed by Bao et al. (2024), has demonstrated superior predictive performance, surpassing GPs models in the three real-world datasets considered. This model was implemented following the specifications of Bao et al. (2024), with code available at github[2].

Approaches I and II do not inherently provide predicted conditional distributions. However, as they optimize the MSE metric, they implicitly learn a Maximum Likelihood Estimate (MLE) of a Gaussian model. Thus, the predictive distribution considered for these approaches was a Gaussian distribution centered on the point prediction with variance equal to the MSE of the validation set. For computational simplicity in the experiments, instead of calculating $\bar{y}_B$ for each batch, we pre-calculated $\bar{y}$ using the entire training set.

**Performance metrics:** We evaluate predictive accuracy using Mean Squared Error (MSE) and Mean Absolute Error (MAE). To assess calibration of the predictive distributions, we report the SQR loss and calibration metric introduced by Kuleshov et al. (2018): calibration $= \sum_{j=1}^{m} \left( \tau^j - \frac{1}{n} \sum_{i=1}^{n} 1 \left[ y_i \leq \hat{q}_i(\tau^j) \right] \right)^2$. For quantile predictions of a calibrated model for a given $\tau$, the proportion of observed values less than or equal to the predicted quantile should approximate $\tau$. While informative, the calibration metric should be interpreted with caution: (i) it represents a coarse global average, meaning that regional over- and underestimations can cancel each other out; and (ii) it is highly sensitive to random variation, as evidenced by the large standard deviations reported in Table 2. Therefore, we regard SQR as the more reliable metric for assessing uncertainty quantification.

## 4.2 California Housing

This dataset comprises pricing information for 20,640 census block groups in California, recorded during the 1990 U.S. census (Pace & Barry, 1997). The main objective is a regression task: predict housing prices, $y$, through the incorporation of six predictive features, $\boldsymbol{x}$, and geographical coordinates, $\boldsymbol{c}$. The predictive features are neighborhood income, house age, number of rooms, number of bedrooms, occupancy and population. All models were trained and evaluated using 80% of the data for training, 10% for validation, and 10% for testing. In the case of SMACNP, to adhere to the specifications of Bao et al. (2024), a training subsample was extracted to represent 10% of the entire dataset.

---

[1]https://github.com/konstantinklemmer/pe-gnn
[2]https://github.com/bll744958765/SMACNP

Table 2: Performance metrics on the California Housing test set. Values are reported as mean $\pm t_{0.975,9} \times \mathrm{SD}/\sqrt{10}$, yielding approximate 95% confidence intervals from the $t$-distribution with nine degrees of freedom. All metrics except # *Param.* and *Time* are multiplied by 100. Boldfaced entries represent the best mean value. The training *Time* is measured in minutes and *Cover.* refers to the observed coverage for 95% confidence intervals.

| Model | # Param. | Time | MSE | MAE | SQR | Cover. (%) | Calibr. |
|---|---|---|---|---|---|---|---|
| GCN | **1,313** | $40.53 \pm 18.77$ | $2.42 \pm 0.19$ | $11.39 \pm 0.40$ | $4.19 \pm 0.15$ | $93.11 \pm 0.46$ | $34.60 \pm 5.44$ |
| PE-GCN $\lambda = $ best | 24,129 | $17.48 \pm 2.81$ | $1.75 \pm 0.05$ | $9.34 \pm 0.10$ | $3.49 \pm 0.03$ | $93.02 \pm 0.55$ | $40.06 \pm 9.91$ |
| PE-GCN (with SQR) | 25,217 | $\mathbf{12.64} \pm 1.90$ | $1.79 \pm 0.06$ | $9.20 \pm 0.18$ | $3.38 \pm 0.07$ | $85.70 \pm 0.85$ | $\mathbf{18.08} \pm 6.02$ |
| PE-GQCN (ours) | 26,201 | $13.86 \pm 0.82$ | $\mathbf{1.11} \pm 0.02$ | $\mathbf{6.83} \pm 0.10$ | $\mathbf{2.61} \pm 0.06$ | $\mathbf{94.47} \pm 1.31$ | $19.62 \pm 17.94$ |
| GAT | **1,441** | $31.93 \pm 10.31$ | $2.40 \pm 0.15$ | $11.32 \pm 0.34$ | $4.17 \pm 0.13$ | $92.95 \pm 0.53$ | $33.72 \pm 6.85$ |
| PE-GAT $\lambda = $ best | 24,290 | $26.62 \pm 4.38$ | $1.77 \pm 0.05$ | $9.36 \pm 0.15$ | $3.54 \pm 0.07$ | $92.98 \pm 0.47$ | $41.32 \pm 7.69$ |
| PE-GAT (with SQR) | 25,345 | $15.03 \pm 2.72$ | $1.84 \pm 0.07$ | $9.34 \pm 0.15$ | $3.42 \pm 0.05$ | $85.34 \pm 1.02$ | $19.85 \pm 5.06$ |
| PE-GQAT (ours) | 26,329 | $\mathbf{14.39} \pm 1.15$ | $\mathbf{1.11} \pm 0.03$ | $\mathbf{6.77} \pm 0.10$ | $\mathbf{2.60} \pm 0.05$ | $\mathbf{95.24} \pm 0.89$ | $\mathbf{11.51} \pm 6.76$ |
| GSAGE | **2,529** | $48.33 \pm 5.07$ | $1.62 \pm 0.06$ | $9.24 \pm 0.20$ | $3.40 \pm 0.08$ | $93.88 \pm 0.31$ | $37.55 \pm 5.84$ |
| PE-GSAGE $\lambda = $ best | 27,426 | $29.37 \pm 4.70$ | $1.07 \pm 0.03$ | $7.21 \pm 0.18$ | $2.73 \pm 0.05$ | $93.94 \pm 0.43$ | $39.83 \pm 9.23$ |
| PE-GSAGE (with SQR) | 28,481 | $16.26 \pm 2.48$ | $1.14 \pm 0.12$ | $7.23 \pm 0.30$ | $2.66 \pm 0.12$ | $85.50 \pm 1.87$ | $28.99 \pm 13.97$ |
| PE-GQSAGE (ours) | 27,417 | $\mathbf{14.99} \pm 1.26$ | $\mathbf{0.85} \pm 0.03$ | $\mathbf{5.93} \pm 0.08$ | $\mathbf{2.26} \pm 0.05$ | $\mathbf{94.46} \pm 1.04$ | $\mathbf{14.43} \pm 17.09$ |
| SMACNP | 748,482 | $73.46 \pm 7.49$ | $2.35 \pm 1.65$ | $8.57 \pm 0.58$ | $4.64 \pm 0.16$ | $99.84 \pm 0.07$ | $288.93 \pm 24.84$ |

As shown in Table 2, **PE-GQNN** delivers strong performance across all evaluation metrics, significantly outperforming traditional GNNs, PE-GNN, and SMACNP. Among models with GSAGE layers — the overall top performers — PE-GQSAGE achieved the lowest values for Time, MSE, MAE, SQR, coverage and calibration error. Specifically, compared to PE-GSAGE, it reduced the training time by 49%, MSE by 21%, MAE by 18%, SQR by 17%, and calibration error by 63%. In contrast, a naive combination of PE-GNN with SQR offered no meaningful improvement over PE-GNN alone.

Figure 2 presents plots that elucidate the behavior of the PE-GQSAGE predictions. The validation MSE curves for a given run are shown in Figure 2a. Panel 2b illustrates the predicted density of a subsample of three observations from the test set. Parametric models typically assume a fixed output structure — such as a Gaussian distribution — which can restrict their expressiveness. In contrast, **PE-GQNN** imposes minimal assumptions on the form of the predictive distribution, offering greater flexibility. As illustrated in Figure 2b, **PE-GQNN** is capable of producing predictive distributions with varying shapes and scales, whereas PE-GNN can only differentiate samples based on their location (mean value). It is worth noting that, in this example, assuming Gaussian distributions is reasonable. However, in scenarios involving asymmetric or multimodal distributions, the flexibility of our model-free approach becomes even more advantageous.

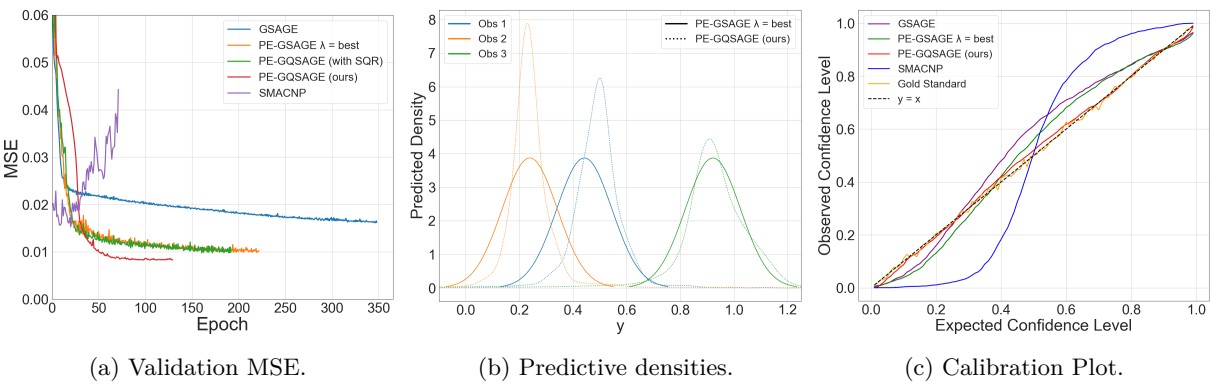

(a) Validation MSE.     (b) Predictive densities.     (c) Calibration Plot.

Figure 2: Diagnostics for the California Housing dataset. All plots refer to a single run of each model. (a) Validation MSE curves. (b) PE-GQSAGE predicted densities of 3 observations of the test set. (c) Calibration Plot.

Figure 2c displays the observed confidence level (averaged across ten runs) for the test set quantile predictions using each of the $m = 99$ $\tau$ values in $[0.01, 0.02, \ldots, 0.99]^\top$. This type of plot was proposed by Kuleshov et al.

(2018). The closer a model gets to the dashed diagonal line, the closer the expected and observed confidence levels are. The Gold Standard represents one Monte Carlo draw from a perfectly specified model, where for each quantile level, the observed confidence level is the observed success rate in $n$ Bernoulli trials with a success probability of $\tau$, where $n$ is the number of test set instances. It is evident that **PE-GQSAGE** has the best calibration performance. This is particularly notable when compared to SMACNP, which exhibits substantial calibration deficiencies due to its tendency to overestimate the variance component.

**Mini-batch size:** As summarized in Figure 1, both PE-GNN and PE-GQNN reconstruct the graph for each mini-batch. According to Klemmer et al. (2023), "*This approach brings a unique advantage: When training with (randomly shuffled) batches, points may have different neighbors in different training iterations (...) This forces PE to learn generalizable features.*" While this finding is generally supported by their experimental analysis, the additional stochasticity introduced by graph reconstruction can degrade performance when mini-batches are too small, as shown in Table 3. Conversely, using very large mini-batches can lead to a substantial increase in computational time. As usual, it is good practice to tune this and other hyperparameters using an efficient optimization framework.

Table 3: Performance metrics on the California Housing test set for different mini-batch sizes. Values are reported as mean $\pm t_{0.975,9} \times \mathrm{SD}/\sqrt{10}$. All metrics except *# Param.* and *Time* are multiplied by 100.

| Batch Size | # Param. | Time | MSE | MAE | SQR | Cover. (%) | Calibr. |
|---|---|---|---|---|---|---|---|
| 128 | 27,417 | **4.73** $\pm 0.45$ | 0.96 $\pm 0.08$ | 6.47 $\pm 0.29$ | 2.46 $\pm 0.10$ | 95.55 $\pm 0.96$ | 83.45 $\pm 54.06$ |
| 256 | 27,417 | 4.76 $\pm 0.33$ | 0.90 $\pm 0.03$ | 6.18 $\pm 0.10$ | 2.35 $\pm 0.05$ | 95.83 $\pm 0.70$ | 36.71 $\pm 16.88$ |
| 512 | 27,417 | 6.45 $\pm 0.54$ | 0.89 $\pm 0.04$ | 6.13 $\pm 0.14$ | 2.31 $\pm 0.07$ | 95.54 $\pm 0.82$ | 39.77 $\pm 18.61$ |
| 1024 | 27,417 | 9.43 $\pm 1.02$ | 0.86 $\pm 0.02$ | 6.02 $\pm 0.12$ | 2.29 $\pm 0.03$ | **95.17** $\pm 0.70$ | 20.37 $\pm 22.54$ |
| 2048 | 27,417 | 14.99 $\pm 1.25$ | 0.85 $\pm 0.02$ | 5.93 $\pm 0.08$ | 2.26 $\pm 0.05$ | 94.46 $\pm 1.04$ | 14.43 $\pm 17.10$ |
| 5096 | 27,417 | 30.01 $\pm 2.11$ | **0.84** $\pm 0.02$ | **5.87** $\pm 0.09$ | **2.23** $\pm 0.05$ | 94.80 $\pm 1.18$ | **8.41** $\pm 6.18$ |

### 4.3 All datasets

Experiments were also conducted on three additional geographic datasets, two of which were previously used by Klemmer et al. (2023) and Bao et al. (2024). See Table 4. The Air Temperature dataset (Hooker et al. (2018)) contains geographical coordinates for 3,076 meteorological stations worldwide, with the goal of predicting mean temperatures ($y$) using mean precipitation levels ($x$). Models were trained with 80% of the data, with 10% for validation and testing each, while SMACNP used a 30% subsample for training, following the specifications of Bao et al. (2024). The 3D Road dataset (Kaul et al. (2013)), includes 434,873 points with latitude, longitude, and altitude for the Jutland, Denmark road network. The task is to interpolate altitude ($y$) using latitude and longitude ($\boldsymbol{c}$). The data were split into 90% for training, 1% for validation, and 9% for testing. The 2021 Australian Census data were used to predict median total personal income (AUD/weekly) across Statistical Areas Level 1 (SA1) based on other census variables. The "2021 General Community Profile for SA1" dataset is publicly available[3], and spatial coordinates were obtained from the `absmapsdata` R package (Mackey, 2025). Tables containing additional income information (numbers 17, 32, 33, 38, 40, 57, 58, and 59), as well as columns with more than 90% zero values, were excluded. The resulting design matrix includes 5,978 features, providing a suitable benchmark for assessing model performance in high-dimensional settings. The dataset was split into 80/10/10% training, validation, and test sets.

Table 5 showcases the experimental results obtained from all four datasets. The **PE-GQNN** models incorporate all innovations discussed in Section 3. **PE-GQNN** outperforms both traditional GNN and PE-GNN. Across all datasets, the innovations introduced by **PE-GQNN** result in substantial reductions in MSE, MAE, and SQR. In the California Housing dataset, **PE-GQNN** consistently outperforms SMACNP in predictive accuracy and provides enhanced uncertainty quantification across all types of GNN layers. Conversely, for the Air Temperature dataset, SMACNP achieves the lowest MSE and MAE but suffers from significantly uncalibrated predictions, reflected by a much higher SQR and calibration error compared to **PE-GQNN**.

---

[3]https://www.abs.gov.au/census/find-census-data/datapacks?release=2021&product=GCP&geography=SA1&header=S

Table 4: Summary of benchmark datasets. Column *Features* refers to the number of columns of the Design Matrix (not including the geographical features Latitude and Longitude). The remaining columns indicate the number of samples in the respective sets.

| Dataset | Features | Training | Validation | Test | Total |
|---|---|---|---|---|---|
| California Housing | 6 | 16,512 | 2,064 | 2,064 | 20,640 |
| Air Temperature | 1 | 2,460 | 308 | 308 | 3,076 |
| 3D Road | 0 | 391,385 | 4,348 | 39,140 | 434,873 |
| Australian Census | 5,978 | 48,471 | 6,059 | 6,059 | 60,589 |

Table 5: Performance metrics from four real-world datasets. In each line, the reported values refer to the implementation with the best performing kind of layer from GCN, GAT and GSAGE. Values are reported as mean $\pm t_{0.975,9} \times \mathrm{SD}/\sqrt{10}$. SMACNP metrics are not reported on the 3D Road and the Australian Census datasets due to high computational costs.

| Dataset/Model | # Param. | Time | MSE | MAE | SQR | Cover. (%) | Calibr. |
|---|---|---|---|---|---|---|---|
| **California Housing** | | | | | | | |
| GNN | 2,529 | 48.33 $\pm 5.07$ | 1.62 $\pm 0.06$ | 9.24 $\pm 0.20$ | 3.40 $\pm 0.08$ | 93.88 $\pm 0.31$ | 37.55 $\pm 5.84$ |
| PE-GNN | 27,426 | 29.37 $\pm 4.70$ | 1.07 $\pm 0.03$ | 7.21 $\pm 0.17$ | 2.73 $\pm 0.08$ | 93.94 $\pm 0.43$ | 39.83 $\pm 9.23$ |
| SMACNP | 748,482 | 73.46 $\pm 7.49$ | 2.35 $\pm 1.65$ | 8.57 $\pm 0.58$ | 4.64 $\pm 0.16$ | 99.84 $\pm 0.07$ | 288.93 $\pm 21.51$ |
| PE-GQNN (ours) | 27,417 | **14.99** $\pm 1.09$ | **0.85** $\pm 0.02$ | **5.93** $\pm 0.07$ | **2.26** $\pm 0.04$ | **94.46** $\pm 0.90$ | **14.43** $\pm 14.81$ |
| **Air Temperature** | | | | | | | |
| GNN | 1,281 | **1.06** $\pm 0.06$ | 2.62 $\pm 0.20$ | 12.43 $\pm 0.45$ | 4.49 $\pm 0.20$ | 92.63 $\pm 1.65$ | 65.25 $\pm 16.09$ |
| PE-GNN | 27,106 | 12.83 $\pm 2.21$ | 0.46 $\pm 0.07$ | 5.04 $\pm 0.37$ | 1.89 $\pm 0.16$ | 94.35 $\pm 1.49$ | 124.72 $\pm 59.40$ |
| SMACNP | 744,482 | 23.32 $\pm 7.35$ | **0.18** $\pm 0.02$ | **2.77** $\pm 0.18$ | 3.80 $\pm 0.06$ | 100.00 $\pm 0.00$ | 600.60 $\pm 12.67$ |
| PE-GQNN (ours) | 26,169 | 8.63 $\pm 1.33$ | 0.23 $\pm 0.03$ | 3.08 $\pm 0.14$ | **1.19** $\pm 0.06$ | 92.01 $\pm 2.58$ | **25.29** $\pm 10.18$ |
| **3D Road** | | | | | | | |
| GNN | 1,153 | 7.20 $\pm 0.67$ | 1.71 $\pm 0.01$ | 10.22 $\pm 0.06$ | 3.57 $\pm 0.01$ | 94.36 $\pm 0.21$ | 36.11 $\pm 4.57$ |
| PE-GNN | 27,106 | 87.27 $\pm 8.83$ | 0.30 $\pm 0.01$ | 4.11 $\pm 0.12$ | 1.50 $\pm 0.03$ | **94.89** $\pm 0.23$ | **26.91** $\pm 9.27$ |
| PE-GQNN (ours) | 23,897 | **6.85** $\pm 0.95$ | **0.01** $\pm 0.00$ | **0.59** $\pm 0.03$ | **0.23** $\pm 0.01$ | 93.53 $\pm 1.55$ | 179.81 $\pm 88.10$ |
| **Australian Census** | | | | | | | |
| GNN | 384,737 | 22.75 $\pm 6.35$ | 0.19 $\pm 0.03$ | 2.90 $\pm 0.26$ | 1.08 $\pm 0.08$ | **95.69** $\pm 0.53$ | **51.51** $\pm 16.08$ |
| PE-GNN | 409,634 | **18.89** $\pm 4.63$ | 0.24 $\pm 0.06$ | 3.43 $\pm 0.48$ | 1.26 $\pm 0.16$ | 95.73 $\pm 0.78$ | 52.18 $\pm 18.14$ |
| PE-GQNN (ours) | 409,625 | 33.26 $\pm 6.66$ | **0.16** $\pm 0.03$ | **2.41** $\pm 0.32$ | **0.90** $\pm 0.12$ | 96.15 $\pm 0.95$ | 104.61 $\pm 61.56$ |

Figure 3 illustrates the spatial distribution of test average absolute residuals, aggregated within hexagonal bins of selected methods. The predictions obtained with **PE-GQNN** appear lighter, especially for the 3D Road dataset. As expected, the highest residuals are concentrated in low-density areas.

## 5 Discussion

In this work, we have proposed the Positional Encoder Graph Quantile Neural Network (**PE-GQNN**) as an innovative framework to enhance predictive modeling for geographic data. Through a series of experiments on real-world datasets, we have demonstrated the significant advantages of the **PE-GQNN** over competitive methods. The empirical results underscored the capability of the **PE-GQNN** to achieve lower MSE, MAE, and SQR across all datasets and GNN backbones compared to traditional GNN and PE-GNN. Notably, the **PE-GQNN** demonstrated substantial improvements in predictive accuracy and uncertainty quantification. The **PE-GQNN** framework's ability to provide a full description of the predictive conditional distribution, including quantile predictions and prediction intervals, provides a notable improvement in geospatial machine learning. The **PE-GQNN** provides a solid foundation for future advancements in the field of geospatial machine learning.

**Limitations:** Although the Lipschitz Monotonic Networks (LMN) (Nolte et al., 2023) can approximate all monotonic Lipschitz bounded functions, many common quantile functions do not satisfy this condition. This often occurs because the derivative $\partial q(\tau)/\partial \tau$ tends to $\infty$ as $\tau$ approaches 1, and to $-\infty$ as $\tau$ approaches 0. The practical implications of this limitation when using LMN blocks remain to be fully understood. However, based on our experiments, it does not appear to pose a major concern. LMNs are unlikely to struggle when approximating truncated distributions, where the support is slightly restricted and the derivative $\partial q(\tau)/\partial \tau$

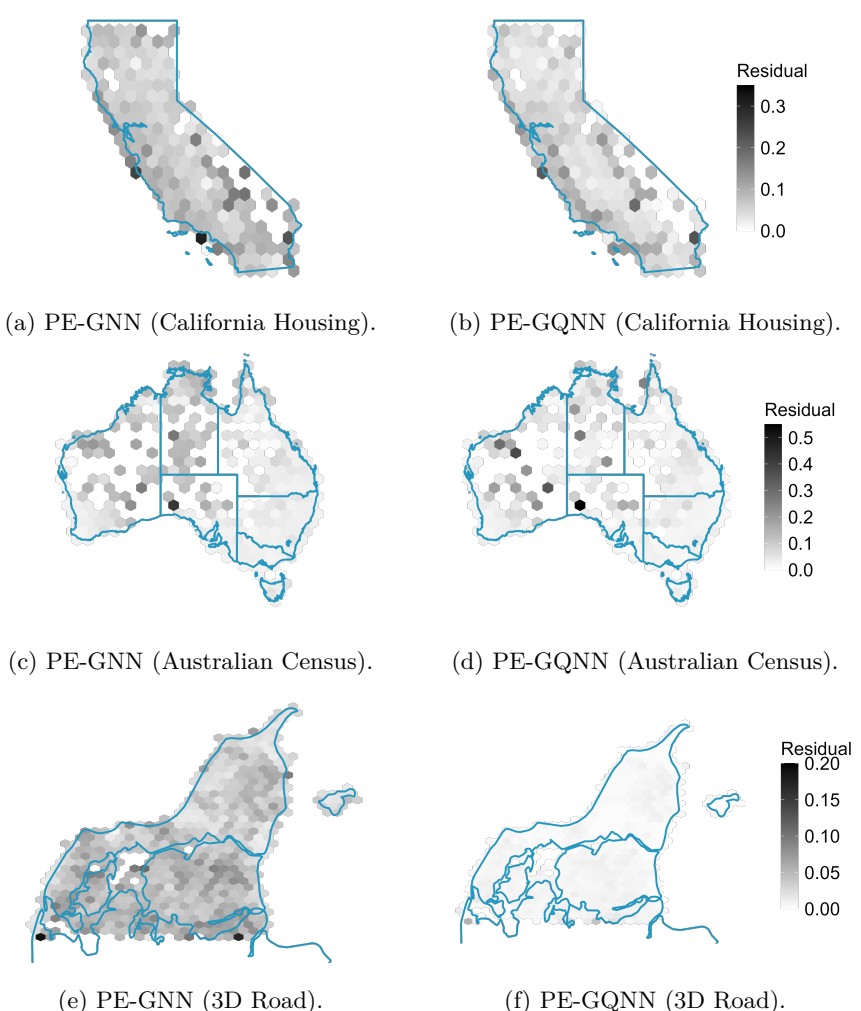

Figure 3: Geographical maps depicting the mean absolute test residuals across ten runs, aggregated within each hexagonal bin.

remains bounded. Naturally, any advances in monotonic neural network architectures would directly enhance the effectiveness of our approach.

## Acknowledgments

Scott Sisson's research was supported by the Australian Research Council. David Nott's research was supported by the Ministry of Education, Singapore, under the Academic Research Fund Tier 2 (MOE-T2EP20123-0009). This study was financed in part by the Coordenação de Aperfeiçoamento de Pessoal de Nível Superior – Brasil (CAPES) – Finance Code 001.

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

# A  Appendix

## A.1  Quantile Neural Networks and recalibration

Figure 4 provides a visual overview of some of the most closely related approaches.

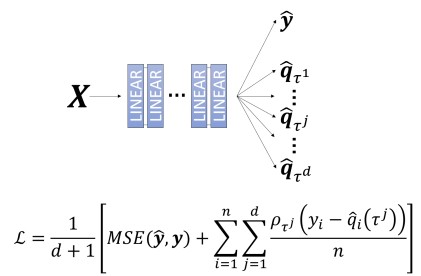

(a) Non-linear quantile regression using NN.  (b) Non-linear multiple quantile regression.

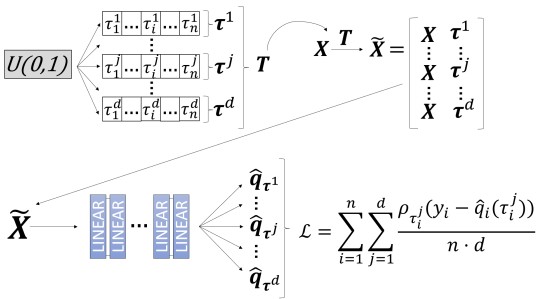
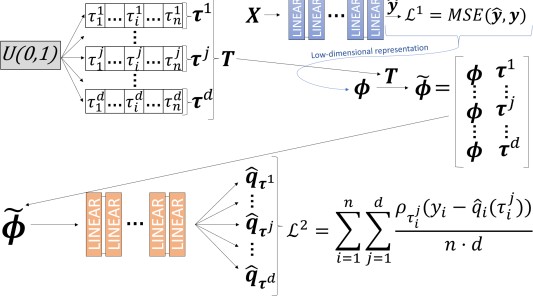

(c) Non-linear quantile function regression.  (d) Two-step density estimation.

Figure 4: (a) For each quantile of interest, a separate NN is trained. (b) Rodrigues & Pereira (2020): one NN outputs $d+1$ predictions: one for the expectation and $d$ for the quantiles. (c) Si et al. (2022): a single NN trained to predict *any* generic quantile of the conditional distribution. (d) Kuleshov & Deshpande (2022): two-step procedure: the first model outputs a low-dimensional representation of the conditional distribution, which a recalibrator then uses to produce calibrated predictions.

### A.2 Positional Encoder

Inspired by the Transformer architecture (Vaswani et al., 2017), Mai et al. (2020) introduced the Positional Encoder (PE) for geographic data. The PE maps the geographic coordinate vector $\boldsymbol{c} = (c_1, c_2) \in \mathbb{R}^2$ of a single datapoint, typically representing latitude and longitude, to a high-dimensional embedding using a set of deterministic sinusoidal transformations followed by a fully connected neural network.

The encoded spatial embedding $\boldsymbol{c}^{\mathrm{emb}} \in \mathbb{R}^d$ is computed as:

$$\boldsymbol{c}^{\mathrm{emb}} = PE(\boldsymbol{c}, \sigma_{\min}, \sigma_{\max}, \Theta_{\mathrm{PE}}) = NN(ST(\boldsymbol{c}, \sigma_{\min}, \sigma_{\max}), \Theta_{\mathrm{PE}}), \tag{4}$$

where $NN(\cdot, \Theta_{\mathrm{PE}})$ denotes a fully connected neural network with parameters $\Theta_{\mathrm{PE}}$, and $ST(\cdot)$ is a sinusoidal transformation defined by:

$$ST(\boldsymbol{c}, \sigma_{\min}, \sigma_{\max}) = (ST_1(\boldsymbol{c}, \sigma_{\min}, \sigma_{\max}); \ldots; ST_S(\boldsymbol{c}, \sigma_{\min}, \sigma_{\max})), \tag{5}$$

with $\sigma_{\min}$ and $\sigma_{\max}$ as hyperparameters. Each component $ST_j(\boldsymbol{c}, \sigma_{\min}, \sigma_{\max})$, for $j \in \{1, \ldots, S\}$, is given by:

$$ST_j(\boldsymbol{c}, \sigma_{\min}, \sigma_{\max}) = \left( \sin\left(\frac{2\pi c_1}{\sigma_j}\right), \cos\left(\frac{2\pi c_1}{\sigma_j}\right), \sin\left(\frac{2\pi c_2}{\sigma_j}\right), \cos\left(\frac{2\pi c_2}{\sigma_j}\right) \right), \tag{6}$$

where the values $\sigma_1, \ldots, \sigma_S$ form a logarithmically spaced grid between $\sigma_{\min}$ and $\sigma_{\max}$, computed as:

$$\sigma_j = \sigma_{\min} \cdot \left(\frac{\sigma_{\max}}{\sigma_{\min}}\right)^{\frac{j-1}{S-1}}. \tag{7}$$

The result of $ST(\boldsymbol{c}, \cdot)$ is a $4S$-dimensional vector that encodes spatial patterns at multiple scales. This vector is then passed through the neural network $NN$ to produce the final embedding $\boldsymbol{c}^{\mathrm{emb}}$.

### A.3 Architectural details

To facilitate a clear and detailed comparison between **PE-GQNN** and PE-GNN, we present the architectures for the California Housing dataset in Tables 6 and 7. The same architectures are used for the Air Temperature dataset, differing only in the number of node features ($p = 1$). For the 3D Road dataset, which is an interpolation task without node features, the GNN block is omitted.

Table 6: Architecture of PE-GQSAGE applied to the California Housing dataset, illustrating input/output dimensions, parameter counts, layer-level annotations, and activation functions.

| Layer (type & shape) | # Param. | Notes | Activation |
|---|---|---|---|
| **GNN block**: $6 \rightarrow 32$ | – | – | – |
|   Input: node features (dim = 6) | – | Node features input | – |
|   SAGEConv (hidden): $6 \rightarrow 32$ | – | GraphConv layer | – |
|     Aggregation | 224 | GSAGE internal layer | – |
|     Update | 192 | GSAGE internal layer | ReLU |
|   SAGEConv (out): $32 \rightarrow 32$ | – | GraphConv layer | – |
|     Aggregation | 1,056 | GSAGE internal layer | – |
|     Update | 1,024 | GSAGE internal layer | ReLU |
| **Positional encoder (PE) block**: $2 \rightarrow 64$ | – | – | – |
|   Input: coordinates (dim = 2) | – | Spatial coordinates input | – |
|   Sinusoidal Transformation (ST): $2 \rightarrow 64$ | 0 | Fourier feature mapping | – |
|   Dropout regularization | – | Dropout rate: $p = 0.5$ | – |
|   Linear (hidden): $64 \rightarrow 128$ | 8,320 | Feedforward layer | ReLU |
|   Linear (hidden): $128 \rightarrow 64$ | 8,256 | Feedforward layer | Tanh |
|   Linear (hidden): $64 \rightarrow 32$ | 2,080 | Feedforward layer | Tanh |
|   Linear (out): $32 \rightarrow 64$ | 2,112 | Feedforward layer | Identity |
| **Concatenation**: GNN (32) + PE (64) | 0 | Combined vectors | – |
| **Fully-connected (FC) block**: $96 \rightarrow 8$ | – | Dimension reduction block | – |
|   Linear (hidden): $96 \rightarrow 32$ | 3,104 | Feedforward layer | Tanh |
|   Linear (hidden): $32 \rightarrow 16$ | 528 | Feedforward layer | Tanh |
|   Linear (out): $16 \rightarrow 8$ | 136 | Feedforward layer | Identity |
| **Concatenation**: FC (8) + $\bar{y}$ (1) + $\tau$ (1) | 0 | Combined vectors | – |
| **Monotonic Layers**: $10 \rightarrow 1$ | – | Final quantile regressor | – |
|   Lipschitz Linear (hidden): $10 \rightarrow 32$ | 352 | Monotonic layer 1 | GroupSort(2) |
|   Lipschitz Linear (out): $32 \rightarrow 1$ | 33 | Monotonic layer 2 | Identity |
| **Total** | 27,417 | All trainable parameters | – |

Table 7: Architecture of PE-GSAGE. Positional encodings derived from spatial coordinates are concatenated with node features prior to the GNN layers. This architecture follows the specification introduced by Klemmer et al. (2023).

| Layer (type & shape) | # Param. | Notes | Activation |
|---|---|---|---|
| **Positional Encoder (PE) block**: $2 \rightarrow 64$ | – | – | – |
|   Input: coordinates (dim = 2) | – | Spatial coordinates input | – |
|   Sinusoidal Transformation (ST): $2 \rightarrow 64$ | 0 | Fourier feature mapping | – |
|   Dropout regularization | – | Dropout rate: $p = 0.5$ | – |
|   Linear (hidden): $64 \rightarrow 128$ | 8,320 | Feedforward layer | ReLU |
|   Linear (hidden): $128 \rightarrow 64$ | 8,256 | Feedforward layer | Tanh |
|   Linear (hidden): $64 \rightarrow 32$ | 2,080 | Feedforward layer | Tanh |
|   Linear (out): $32 \rightarrow 64$ | 2,112 | Feedforward layer | Identity |
| **Concatenation**: PE (64) + Features (6) | 0 | Combined input vector | – |
| **GNN block**: $70 \rightarrow 32$ | – | – | – |
|   Input: node features (dim = 70) | – | PE + Node attributes | – |
|   SAGEConv (hidden): $70 \rightarrow 32$ | – | GraphConv layer | – |
|     Aggregation Linear | 2,272 | GSAGE internal layer | – |
|     Update Linear | 2,240 | GSAGE internal layer | ReLU |
|   SAGEConv (out): $32 \rightarrow 32$ | – | GraphConv layer | – |
|     Aggregation Linear | 1,056 | GSAGE internal layer | – |
|     Update Linear | 1,024 | GSAGE internal layer | ReLU |
| **Fully-connected layers**: $32 \rightarrow 1$ | – | – | – |
|   Linear (to target): $32 \rightarrow 1$ | 33 | Final output layer | Identity |
|   Linear (to Moran's I): $32 \rightarrow 1$ | 33 | Auxiliary output layer | Identity |
| **Total** | 27,426 | All trainable parameters | – |

