# OpenReview forum: "Positional Encoder Graph Quantile Neural Networks for Geographic Data"
_TMLR — Accepted by TMLR_

### Review · Reviewer_q7n9 · 2025-10-06

**Summary Of Contributions:**

In this paper, the authors introduced the Positional Encoder Graph Quantile Neural Network (PE-GQNN), a novel framework designed to improve predictive modeling for geographic data. The main contributions of the study are as follows: Experimental results demonstrated that PE-GQNN consistently outperformed traditional GNN and PE-GNN models, achieving lower values in MSE and MAE metrics. Furthermore, the model showed significant gains in both predictive accuracy and the quality of uncertainty quantification.

**Additional Comments:**

None

**Audience:**

Yes

**Audience Explanation:**

.

**Claims And Evidence:**

Yes

**Claims Explanation:**

**Strengths** :

- the proposed method combines both quantile prediction and distribution calibration in a single model for more efficiency
- the proposed method allows for good predictions accross any quantile level
- the experiments accross multiple dataset are convincing

**Weaknesses** :

- The technical contribution of the paper is limited. Indeed, the proposed method is essentially a combination of PE-GNN and quantile regression. Techniques such as response variables and quantile parameters into neural networks are included. However, these contributions are incremental rather than fundamentally novel.

- The paper does not address the computational cost associated with applying the method to high-dimensional data, which could be a significant limitation in real-world scenarios.

**Requested Changes:**

See weaknesses section.

---

> ### Author Response · Authors · 2025-10-29
> **Response to Reviewer q7n9**
>
> We sincerely thank the reviewer for their supportive and constructive feedback.
>
> **Response to Comment 1**
>
> Our approach indeed builds upon well-established techniques, such as Simultaneous Quantile Regression (SQR) and PE-GNN. However, our analysis using the California Housing dataset revealed that a straightforward combination of these methods (denoted as PE-GSAGE (with SQR) in Table 2) did not yield any improvement over PE-GNN alone. This indicates that the additional methodological innovations, particularly the way the quantile level $\tau$ is incorporated into the network (through a transformed representation and closer to the output), were crucial to making the method effective. To the best of our knowledge, this treatment of $\tau$ is novel within the quantile regression literature. Moreover, we have shown analytically that, somewhat surprisingly, our formulation recovers Gaussian regression as a special case, despite the distinct nature of the associated loss functions (MSE vs. SQR). This result opens new directions for developing more flexible and theoretically grounded quantile regression models.
>
> **Response to Comment 2**
>
> We appreciate the reviewer’s insightful suggestion. There were theoretical reasons to expect that our method would perform well on datasets with a high number of features, but none of the datasets originally used in the PE-GNN study were appropriate to assess this hypothesis. In response to the reviewer’s comment, we have now included a fourth benchmark dataset; predicting median per-capita income across Australia’s Statistical Areas (SA1) using other census variables. This dataset includes a design matrix with 5,978 columns, providing a suitable setting to assess high-dimensional performance. The results were strong, with PE-GQNN achieving higher predictive performance. Across all experiments, PE-GQNN exhibited computational times comparable to PE-GNN (sometimes better, sometimes worse), and consistently much lower than SMACNP. These new results are presented in the expanded Table 3, which now reports computational times and additional performance metrics for all experiments.

---

### Review · Reviewer_aaBq · 2025-10-08

**Summary Of Contributions:**

The paper proposes a few modifications to the PE-GNN method to enhance its ability to make accurate spatial predictions and to quantify uncertainty. The following modifications are introduced:
(1) a new method fro quantile regression task that merges the two-step procedure of Kuleshov & Deshpande (2022) into a single model by postponing the concatenation of the τ value. τ is transformed beforehand to facilitate learning and provide better
control over the form of the predicted distribution. This approach aims to
enhances the network’s ability to model uncertainty and to make it more robust to high-dimensional
predictor spaces.
(2) the proposed approach modifies the way the nodes’ features and the spatial embeddings are concatenated in PE-GNN.

**Additional Comments:**

In the introduction, the contributions are quite wordy and not easy to follow

**Audience:**

Yes

**Audience Explanation:**

Making geospatial predictions more robust is of interest to geospatial and, potentially, machine learning audience. The performance  improvement is not significant, though.

**Broader Impact Concerns:**

I do not see major ethical implications of this work

**Claims And Evidence:**

Yes

**Claims Explanation:**

The proposed approach teases on 3 geospatial datasets and the performance is compared to current GNN approaches (GNNs, PE-GNN, SMACNP and GSAGE). The approach demonstrates some improvement on a variety of metrics, including running time, MSE, MAE etc.

**Requested Changes:**

As with many geospatial datasets, the visualisation of the results might be useful. Figure 4 in appendix might be brought up in the main paper (or even better, it can be dome for one of the other datasets). However, the captions should clearly describe what does the colour represent. Additionally, it'll be good to demonstrate not the predictions themselves, but the difference between real and predicted values: it will make the improvement in the model performance more obvious.

---

> ### Author Response · Authors · 2025-10-29
> **Response to Reviewer aaBq**
>
> We thank the reviewer for their fair comments and for sharing our view on the importance of developing more robust models for geospatial regression.
>
> **Response to Comment 1**
>
> Regarding the performance of our approach, please note that in all datasets, the response variable was normalized during preprocessing, consistent with the procedure adopted in the original PE-GNN paper. Consequently, the magnitude of the error metrics (e.g., MSE) is naturally small. Nevertheless, our method achieved a reduction of at least 15\% in MSE across all benchmark datasets. In high-stakes applications, even such seemingly modest improvements can translate into substantial practical or financial gains. Similarly, reductions in quantile estimation errors (captured by the SQR metric) can also lead to meaningfully better decision outcomes.
>
> **Response to Comment 2**
>
> We appreciate the reviewer’s valuable suggestions, which have significantly improved the clarity and presentation of our results. In particular:
> all plots have been moved to the main paper;
> new plots for two other datasets were included;
> the figure captions were revised to clearly state how the maps should be interpreted; and
> the color scale now represents the average absolute error within each hexagonal bin, reducing overplotting and improving visual interpretability.
>
> **Response to Comment 3**
>
> We agree that the original paragraph describing the paper’s contributions was overly verbose and difficult to follow. We have rewritten this part to make it more concise, retaining only the essential information and relocating secondary details to later sections.

---

### Review · Reviewer_kFfa · 2025-10-13

**Summary Of Contributions:**

The paper introduces the Positional Encoder Graph Quantile Neural Network (PE-GQNN), which integrates positional-embedding-aware graph neural networks with quantile regression to enable uncertainty-aware spatial prediction. Building on prior work, the approach removes the need for post-hoc calibration by combining prediction and uncertainty estimation within a single end-to-end framework. The method appears robust in high-dimensional predictor spaces and produces well-calibrated predictive distributions. Empirical results across several real-world datasets show that PE-GQNN achieves strong and often superior performance compared to established baselines across multiple evaluation metrics.

**Additional Comments:**

Some references should be added for specific arguments in the paper: for example, the introduction states “...poorly calibrated intervals, such as 80% intervals that fail to contain the true outcome 80% of the time,” but no citation is provided.

**Audience:**

Yes

**Audience Explanation:**

Despite some limitations in the experimental setup, this paper represents a strong contribution to the development of GNN-based methods for geographical data. The integration of quantile neural networks with positional encoders is novel and adds clear value to the graph-ML community. By producing full predictive distributions rather than single point estimates, the proposed model enables uncertainty-aware spatial prediction, an aspect still uncommon in graph-based frameworks. Although primarily motivated by spatial applications, the framework’s quantile-based uncertainty modeling and monotonic design principles could, in principle, extend to non-spatial or relational graph problems, underscoring its potential broader impact.

**Claims And Evidence:**

No

**Claims Explanation:**

The paper is well-written and presents an interesting contribution, but there are several concerns regarding the evidence provided and some methodological choices.

First, while the paper emphasizes the scalability of the proposed components, there is no theoretical analysis of the model’s computational complexity. Table 2 provides runtime estimates for a single dataset, but without explicit details on dataset sizes, it is difficult to assess the actual computational cost or scalability limits of the approach.

Second, the claims regarding uncertainty quantification are largely theoretical. The experimental evaluation provides limited evidence that the model delivers reliable or calibrated uncertainty estimates in practice.

From a methodological standpoint, there are no theoretical or empirical guarantees of model stability or convergence. The graph is recomputed for every batch, which means the adjacency structure changes dynamically. This introduces additional stochasticity that may not always improve generalization and could affect training stability. Moreover, the impact of batch size on both performance and computational cost is not analyzed, although it is likely to play an important role.

The decision to use a single Monte Carlo sample for τ (quantile level) is also questionable. While the law of large numbers offers theoretical justification, it would be important to examine in practice how this choice affects the variance of gradient estimates and the stability of training.

Finally, although Lipschitz Monotonic Networks are introduced to prevent quantile crossing, these constraints may restrict model flexibility and still yield only approximately monotonic quantiles. A deeper empirical investigation of this trade-off would strengthen the work.

Overall, the robustness analysis is insufficient to fully validate the proposed method. Performance results should be reported as averages over multiple runs with associated error estimates to provide a more reliable assessment of model consistency and generalization.

**Requested Changes:**

Please see above in the claims and evidence section.

---

> ### Author Response · Authors · 2025-10-29
> **Response to Reviewer kFfa**
>
> We sincerely thank the reviewer for carefully reading our manuscript and for providing detailed feedback that helped us improve the reliability of our assessment of model performance and scalability.
>
> **Response to Comment 1**
> We agree with the reviewer. Computational time depends on several factors, including the size and shape of the training dataset, the batch size, the learning rate, and the hyperparameters of the early stopping regularizer. Based on our experiments, we understand that the model can be applied to problems with at least a few million rows and several thousand features on a standard user-grade PC. We now explicitly report the computational times for all datasets. Although dataset sizes were approximately mentioned in the text, we have added the exact number of observations and features to Table 4 for clarity and convenience.
>
> **Response to Comment 2**
> We agree that, as with any “inherently calibrated” or “recalibrated” model, our approach may produce unreliable uncertainty quantification in some applications. In practice, uncertainty-aware methods aim to achieve better-calibrated (rather than perfectly calibrated) uncertainty estimates relative to baseline approaches. We have clarified this distinction in the revised text.
> Empirically, our models achieved reductions of at least 14\% in the test SQR metric across all datasets, supporting that the proposed theoretical refinements improve calibration in practice.
>
> While our method performs strongly under SQR, it performs more variably under Calibr. However, in general, the Calibr. metric may be misleading for two reasons: (i) it is a coarse global average; so regional over- and under-estimations can cancel out, similarly to how uniformly distributed marginal PIT values do not necessarily imply calibration; and (ii) it is highly sensitive to random variation (as illustrated in the new sensitivity analysis in Table 3). As a result, we feel that SQR is the more reliable metric here for assessing uncertainty quantification.
>
> **Response to Comment 3**
>
> The idea of recomputing the graph for each batch was originally introduced by Klemer (2023), who noted that “This approach brings a unique advantage: When training with (randomly shuffled) batches, points may have different neighbors in different training iterations (…) This forces PE to learn generalizable features.” Still, we agree with the reviewer that this additional stochasticity can potentially degrade performance for small mini-batch sizes. This behavior is confirmed in our new sensitivity analysis (Table 3). We thank the reviewer for highlighting this issue, as we had adopted the same batch size as in PE-GNN without providing a sensitivity analysis. We have also added a short discussion on this point.
>
>
> **Response to Comment 4**
>
> Following Tagasovska \& Lopez-Paz (2019), we sample a single quantile level $\tau$ per training point and mini-batch during training. The number of terms in the Monte Carlo approximation of the gradients is therefore given by $n_B$. Because the batch size should not be too small (as discussed in Response to Comment 3), each gradient estimate considers losses over more than a thousand different quantile levels in all our experiments. This design produced smooth and stable training curves, as shown in Figure 2a. We have reorganized this discussion in the manuscript to provide a clearer justification for our design choice.
>
> **Response to Comment 5**
>
> We agree that there is a trade-off between model capacity and probabilistic coherence. The decision to incorporate monotonic layers is problem-dependent. In practice, it is straightforward to assess on a validation set whether quantile crossing is significant. When it is not (especially since SQR (Tagasovska \& Lopez-Paz, 2019) already mitigates this issue), adding monotonic layers may be unnecessary or even counterproductive. As usual, we recommend exploring multiple network configurations during hyperparameter tuning to achieve a good balance between expressiveness and coherence. This important practical discussion has been incorporated into Section 3.
>
> **Response to Comment 6**
>
> We agree that averaging results over multiple runs and reporting empirical standard deviations is the most appropriate way to mitigate unwanted random variation in performance metrics. Due to computational constraints, we are currently rerunning all experiments with 10 replications each. We kindly inform the reviewer that the manuscript will be updated with these more definitive results in the coming days.
>
> **Response to Comment 7**
>
> We appreciate the suggestion and have included additional references to better contextualize our work.

---

### Author Response · Authors · 2025-11-08
**Summary of Main Revisions and Enhancements**

We are grateful to all reviewers for their time and thoughtful feedback, which greatly contributed to improving the quality and clarity of our manuscript.

We are also pleased by the positive evaluations. Reviewer **q7n9** emphasized the value of integrating quantile prediction and distribution calibration within a unified framework and noted the convincing experimental results across multiple datasets. Reviewer **aaBq** recognized that the proposed approach enhances robustness and accuracy in geospatial prediction tasks. Reviewer **kFfa** provided detailed and constructive feedback, while also acknowledging that “*this paper represents a strong contribution to the development of GNN-based methods for geographical data. The integration of quantile neural networks with positional encoders is novel and adds clear value to the graph-ML community.*”


In this revised version, we have made several important improvements to address the reviewers’ comments:

**New high-dimensional benchmark dataset**: Following the suggestion by Reviewers **q7n9** and **kFfa**, we added a fourth benchmark dataset to evaluate the method’s scalability. The new task involves predicting median per-capita income across Australia’s Statistical Areas (SA1) using census data with 5,978 features. The results confirm that PE-GQNN performs strongly in high-dimensional settings.

**Sensitivity analysis of batch size**: In response to Reviewer **kFfa**, we conducted a new sensitivity analysis investigating the impact of batch size on predictive performance and computational time. The results, presented in Table 3, confirm that while small batches introduce instability due to increased stochasticity, moderate-to-large batches yield stable training dynamics.

**Expanded evaluation and reporting of uncertainty**: We now average results over 10 independent runs and report empirical means and standard deviations for all metrics, ensuring a more reliable assessment of model consistency and robustness.

**Improved visualization and interpretation**: Following Reviewer **aaBq**’s recommendation, we moved all key visualizations to the main paper, enhanced figure captions to clearly explain how to interpret the maps, and added new plots for two additional datasets. The color scale now represents the average absolute error within each hexagonal bin, improving interpretability and reducing overplotting.

**Clarified computational complexity and scalability**: We expanded the discussion on runtime performance and now explicitly report the number of observations, features, and computational times for all datasets (see Table 4).

**Refined methodological discussions**: We reorganized and clarified the reasoning behind (i) recomputing the graph at each mini-batch, (ii) using a single Monte Carlo sample for the quantile level, and (iii) incorporating monotonic layers. These discussions now better justify our design choices and highlight the trade-offs between model expressiveness, coherence, and computational efficiency.

Together, these additions and clarifications strengthen the paper’s theoretical justification, empirical reliability, and clarity of presentation.

---

### Decision · Action_Editor_LFY1 · 2025-11-29

**Recommendation:** Accept as is

**Audience:**

Yes

**Audience Explanation:**

For the sake of transparent decision-making, I include here TMLR's criteria for acceptance that each accepted paper must meet:

- Are the claims made in the submission supported by accurate, convincing, and clear evidence?
- Would at least some individuals in TMLR's audience be interested in knowing the findings of this paper?

Reviewers agree that the paper meets the second criterion mentioned above related to "Audience", and some even believe that it goes further by making a meaningful contribution to the development of GNN-based methods for geographical data.

**Claims And Evidence:**

Yes

**Claims Explanation:**

For the sake of transparent decision-making, I include here TMLR's criteria for acceptance that each accepted paper must meet:

- Are the claims made in the submission supported by accurate, convincing, and clear evidence?
- Would at least some individuals in TMLR's audience be interested in knowing the findings of this paper?

All reviewers agree that the paper clearly meets the first criterion related to "Claims And Evidence" by providing claims that are sound and well supported by evidence. The reviewers raised some questions and concerns, which the authors have adequately addressed in the revisions to the reviewers' satisfaction.